# On the Duality between Gradient Transformations and Adapters

Lucas Torroba-Hennigen [1]   Hunter Lang [1]   Han Guo [1]   Yoon Kim [1]

## Abstract

We study memory-efficient optimization of neural networks (in particular language models) with *linear gradient transformations*, where the gradients are linearly mapped to a lower dimensional space than the full parameter space, thus saving memory required for gradient accumulation and optimizer state persistence. The model parameters are updated by first performing an optimization step in the lower dimensional space and then going back into the original parameter space via the linear map's transpose. We show that optimizing the model in this transformed space is equivalent to reparameterizing the original model through a *linear adapter* that additively modifies the model parameters, and then only optimizing the adapter's parameters. When the transformation is Kronecker-factored, this establishes an equivalence between GaLore (Zhao et al., 2024) and one-sided LoRA (Hu et al., 2022). We show that this duality between gradient transformations and adapter-based reparameterizations unifies existing approaches to memory-efficient training and suggests new techniques for improving training efficiency and memory use.

## 1. Introduction

Training neural networks, in particular large language models (LLMs), can be extremely memory-intensive. Standard approaches for LLM training use gradient accumulation across multiple batches and optimizers such as Adam (Kingma & Ba, 2015), which maintains estimates of the first and second moments of the (stochastic) gradient. Hence, the amount of GPU memory needed for standard training can be as much as four times the amount of memory needed to store the model (assuming the gradients/optimization states are kept in the same precision as the model parameters).

In response, a wealth of literature has developed around memory-efficient training methods. Most of these fall into one of two families. The first involves modifications to the model parameterization, in particular by introducing "adapters" to the model architecture that have a small number of additional parameters, and only tuning those adapters (Houlsby et al., 2019; Li & Liang, 2021; Hu et al., 2022). Adapters such as LoRA increase the total number of parameters but reduce the number of trainable parameters, resulting in overall memory savings. While LoRA was originally introduced for memory-efficient *finetuning*, recent works such as ReLoRA (Lialin et al., 2024), LoRA-the-Explorer (LTE; Huh et al., 2024), and Flora (Hao et al., 2024) find that LoRA can even enable memory-efficient *pretraining* if the adapters are periodically merged back into the full model (and then reinitialized).

The second family of methods involves more direct changes to the optimization strategy, either by designing optimizers that store fewer extra bits of information per parameter (Anil et al., 2019; Shazeer & Stern, 2018), or (broadly) compressing the gradients, e.g., via quantization (Bernstein et al., 2018; Dettmers et al., 2022; Li et al., 2024a) or low-rank approximations (Gooneratne et al., 2020; Huang et al., 2023). For LLMs, GaLore (Zhao et al., 2024) has recently emerged as a promising gradient compression approach for memory-efficient pretraining. GaLore transforms the gradient matrices of linear layers via projections derived from an SVD of the gradient matrix, and then performs optimization in this projected space.

Is there a relationship between methods that directly transform/compress gradients, and adapter-based methods that reparameterize the underlying model into frozen and trainable components? In the case of GaLore and LoRA, recent works find that the answer is *yes*, in particular showing that training a LoRA adapter with one side frozen can be seen as a form of gradient compression where the gradient matrices are sketched to a lower dimensional space with random matrices (Hao et al., 2024) or through SVD-based projections (Loeschcke et al., 2024).

In this work, we show that the connection between GaLore and LoRA is more general by proving that training a neural network by applying an *arbitrary linear transformation* to the gradient vector is equivalent (in the sense that the re-

---

[1]Massachusetts Institute of Technology. Correspondence to: Lucas Torroba-Hennigen <lucastor@mit.edu>.

*Proceedings of the 42$^{nd}$ International Conference on Machine Learning*, Vancouver, Canada. PMLR 267, 2025. Copyright 2025 by the author(s).

sulting models are the same and have the same optimization trajectory) to reparameterizing the neural network through a *linear adapter* that additively modifies the original parameters, and then only training the adapter. When applied to (vectorized) matrices with a particular Kronecker factorization of the linear map, our results recover the equivalence between GaLore and one-sided LoRA.

Our empirical experiments study this connection between linear gradient transformations and adapter-based reparameterizations in the context of memory-efficient LLM training. First, we perform a comparison across gradient projection-based and LoRA-based approaches for memory-efficient training and find that randomly sketching gradients works particularly well (§4.1). We also exploit the adapter view of projected-gradient training[1] by developing a QLoRA-style (Dettmers et al., 2023) approach to GaLore-style training. Second, we show that the gradient projection view of LoRA adapters can improve distributed training of LLMs with parallel LoRA adapters (Huh et al., 2024) by suggesting an initialization scheme of worker-specific LoRA adapters tailored for distributed training (§4.2). These results collectively demonstrate that this *duality* between linear gradient transformations and adapter-based reparameterizations is a productive lens with which to view neural network optimization, since it unifies several existing approaches and suggests new techniques for improving training efficiency and performance.

## 2. Background

### 2.1. Memory Characteristics of LLM Training

LLM training makes use of accelerators like GPUs, which requires storing important data in rapidly accessible, on-device memory.[2] The bulk of this memory consumption can be broken down into four main categories.

**Model parameters.** We must keep the model's parameters in memory, since these are used in various stages of the training process (e.g., to compute gradients). Here, it is useful to distinguish *trainable* parameters (which get updated regularly during training) from *non-trainable* parameters (which are not updated during training but may still be used in gradient computation).

**Gradients.** LLMs are trained using (variants of) stochastic gradient descent, which requires an estimate of the gradient of the loss function with respect to each trainable parameter. Standard LLM training uses a large number of samples to estimate the gradient, which necessitates gradient accumulation across multiple mini-batches of data.[3]

**Optimizer states.** In addition to the gradient itself, most optimizers used in LLM training persist other state across steps. Adam (Kingma & Ba, 2015) and AdamW (Loshchilov & Hutter, 2019) maintain running averages of the gradient and the gradient squared (i.e., an estimate of first- and second-order moments), which require two floats per trainable parameter. Examples of techniques that reduce optimizer memory include 8-bit Adam (Dettmers et al., 2022), which stores Adam states in lower precision, and AdaFactor (Shazeer & Stern, 2018), which modifies Adam to use fewer floats per parameter.

**Activations.** LLM gradients are almost always obtained using reverse-mode automatic differentiation (Griewank & Walther, 2008). This consists of building a description of the LLM during a forward pass, in terms of a computation graph of its operations, and storing all (possibly intermediate) results required to subsequently compute the gradients of the neural network. The simplest way to reduce activation memory is by breaking batches into smaller micro-batches and performing more gradient accumulation steps. Other techniques include gradient checkpointing (Chen et al., 2016; Jain et al., 2020), which trades off compute for activation memory by recomputing quantities during the backward pass, and random projections (Bershatsky et al., 2022; Liu et al., 2023), which produce stochastic estimators of gradients based on sketched activations.

This work is mostly concerned with training LLMs in memory-constrained regimes where the model, optimizer, and gradient memory dominate, since activation storage can be made small by, e.g., reducing the microbatch size. As such, we will only focus on those categories in our calculations.

### 2.2. LoRA and GaLore

This paper centers mainly around two memory-efficient training techniques: low-rank adapters (LoRA; Hu et al., 2022) and gradient low-rank projections (GaLore; Zhao et al., 2024). LoRA reparameterizes the model's linear layers as $\mathbf{Y} = (\mathbf{W} + \mathbf{AB})\mathbf{X}$, where $\mathbf{W}$ is the model's original weight matrix and $\mathbf{A}, \mathbf{B}$ are matrices such that $\text{rank}(\mathbf{AB}) < \text{rank}(\mathbf{W})$. $\mathbf{W}$ remains frozen and only $\mathbf{A}, \mathbf{B}$ are optimized; thus, while the total number of model parameters is increased, the number of trainable parameters is decreased, which can lead to memory savings. Recent

---

[1]Note that this notion of performing gradient descent with projections of the gradient is distinct from *projected gradient descent* (PGD) from the optimization literature.

[2]While offloading to CPU is theoretically possible, bandwidth limitations often make this infeasible in practice.

[3]While there are methods that perform an optimizer step as soon as a gradient is estimated (thus eliminating the need to allocate memory for gradient accumulation; e,g., LOMO, Lv et al., 2024), this is not standard in LLM training since it can place restrictions on sequence length: for example GaLore with LOMO only trains on 256-length sequences. We thus train with gradient accumulation in the present work.

works obtain even further memory savings by working with a compressed version of $\mathbf{W}$ (Dettmers et al., 2023; Guo et al., 2024; Li et al., 2024b). While LoRA was originally proposed in the context of memory-efficient finetuning, ReLoRA (Lialin et al., 2024), LoRA-the-Explorer (Huh et al., 2024), and Flora (Hao et al., 2024) show that by periodically merging the low-rank components with the full weights and reinitializing them, LoRA can enable reasonably performant memory-efficient pretraining from scratch.

GaLore provides an alternative approach to memory-efficient pretraining. Instead of reparameterizing the weights to be a combination of full-rank and low-rank matrices—which increases the number of model parameters—GaLore performs a low-rank compression of the *gradient matrix* $\overline{\mathbf{W}}$ instead. The optimizer update is computed in this lower dimensional space, transformed back to the original space, and only then applied to the parameters. Specifically, given a gradient matrix $\overline{\mathbf{W}} \in \mathbb{R}^{m \times n}$, GaLore uses a matrix $\mathbf{P} \in \mathbb{R}^{k \times m}$ (with $k < m$) to transform the gradient via $\mathbf{P}\overline{\mathbf{W}} \in \mathbb{R}^{k \times n}$, feeds this compressed gradient into a regular optimizer to obtain a pseudo-parameter update $\Delta \in \mathbb{R}^{k \times n}$, and then updates the original parameters via $\mathbf{P}^\top \Delta$. In practice, $\mathbf{P}$ is given by the top singular vectors of $\overline{\mathbf{W}}$, where in order to amortize the cost of SVD, $\mathbf{P}$ is updated only every so often. As with LoRA, GaLore reduces the memory needed to store the optimizer states, since optimization happens in the lower dimensional space.

# 3. Duality between Linear Gradient Transformations and Adapters

In this section, we prove that training a neural network using linear transformations of the gradient is equivalent to reparameterizing the neural network using specific linear adapters. We begin with the general case, where all parameters are treated as arbitrary vectors (Thm. 1). We then show how applying a Kronecker-factored linear transformation to the gradients of linear layers of the network is equivalent to training the model with a version of LoRA which inserts a trainable matrix between the LoRA matrices (Prop. 2). From this, we further show that specializing to a specific choice of Kronecker-factored transformation establishes an equivalence between GaLore (Zhao et al., 2024) and "one-sided" LoRA (Hu et al., 2022) where one of the LoRA matrices is initialized in a particular way and kept frozen, while only the other one is trained (Cor. 3); this recovers the equivalence established in recent work (Hao et al., 2024; Loeschcke et al., 2024).

## 3.1. General Case

Let $f(\mathbf{X}; \Theta)$ be a neural network over input $\mathbf{X}$ with trainable parameters $\Theta \in \mathbb{R}^d$, and further let $\overline{\Theta} \in \mathbb{R}^d$ be the gradient of some differentiable loss function $\mathcal{L}$ of the network $f$ with respect to $\Theta$, computed on a random data minibatch. We use the superscript $(\cdot)^{(t)}$ to specify a particular quantity's value after $t$ optimizer steps, e.g., $\Theta^{(t)}$ are the network's parameters after $t$ optimizer steps. We are also interested in the *optimization trajectory* of a model $(\Theta^{(0)}, \Theta^{(1)}, \ldots, \Theta^{(t)})$. Our results show that two optimization trajectories—one from training with linear gradient transformations, and one from training with linear adapters—are equivalent.

Typical approaches to neural network optimization use optimizers that maintain a state $\xi_\Theta^{(t)}$ and obtain $\Theta^{(t+1)}$ via,

$$(\Delta_\Theta^{(t)}, \xi_\Theta^{(t+1)}) = \text{Optimizer}(\overline{\Theta}^{(t)}, \xi_\Theta^{(t)}) \qquad (1)$$

$$\Theta^{(t+1)} = \Theta^{(t)} + \Delta_\Theta^{(t)}. \qquad (2)$$

For example, Adam[4] (Kingma & Ba, 2015) maintains first- and second-moment estimates of the gradient entries in its state $\xi_\Theta^{(t)} = (\boldsymbol{\mu}_\Theta^{(t)}, \boldsymbol{\nu}_\Theta^{(t)})$, and the optimizer update is:

$$\begin{aligned}\mu_{\Theta,i}^{(t+1)} &= (1-\beta_1)\overline{\Theta}_i^{(t)} + \beta_1\mu_{\Theta,i}^{(t)} & \Delta_{\Theta,i}^{(t)} &= -\gamma\frac{\mu_{\Theta,i}^{(t+1)}}{\sqrt{\nu_{\Theta,i}^{(t+1)}+\epsilon}} \\ \nu_{\Theta,i}^{(t+1)} &= (1-\beta_2)(\overline{\Theta}_i^{(t)})^2 + \beta_2\nu_{\Theta,i}^{(t)}\end{aligned}$$

where $\gamma \in \mathbb{R}^+$ is the learning rate, $\beta_1, \beta_2 \in [0, 1)$ control the exponential moving averages of the gradient moments, and $\epsilon$ is present for numerical stability. In this case, the dimensionality of the optimizer states is proportional to the dimensionality of our gradient estimate $\dim(\overline{\Theta}^{(t)}) = d$, i.e., $\Delta_\Theta^{(t+1)} \in \mathbb{R}^d, \xi_\Theta^{(t)} \in \mathbb{R}^{2d}$.

Now consider optimizing $\Theta$ with *linearly transformed gradient dynamics*, where the gradient $\overline{\Theta}$ is mapped to an $r$-dimensional space by a matrix $\mathbf{S} \in \mathbb{R}^{r \times d}$. In this case, we can use the transpose of the linear map to go back into the original parameter space resulting in the following update:

$$(\Delta_{\mathbf{S}\Theta}^{(t)}, \xi_{\mathbf{S}\Theta}^{(t+1)}) = \text{Optimizer}(\mathbf{S}\overline{\Theta}^{(t)}, \xi_{\mathbf{S}\Theta}^{(t)})$$

$$\Theta^{(t+1)} = \Theta^{(t)} + \mathbf{S}^\top \Delta_{\mathbf{S}\Theta}^{(t)},$$

where we have used the subscript $\mathbf{S}\Theta$ to emphasize the fact that the optimizer is now operating on a different space, i.e., as if we were optimizing on $\mathbb{R}^r$, instead of the original parameter space, $\mathbb{R}^d$. For example, if we were using Adam as our optimizer, then this change would cause the dimensionality of the optimizer update and states to be proportional to $r$ instead of $d$, viz., $\Delta_{\mathbf{S}\Theta}^{(t+1)} \in \mathbb{R}^r, \xi_{\mathbf{S}\Theta}^{(t)} \in \mathbb{R}^{2r}$.

Let us further consider a *reparameterization* of the neural network parameters as $f(\mathbf{X}; \Theta + \mathbf{S}^\top \Lambda)$ with $\Lambda \in \mathbb{R}^r$. Specifically, suppose that we keep $\Theta$ and $\mathbf{S}$ fixed, and only

---

[4]We omit bias correction for simplicity; one can easily handle it by adding the current timestep to our optimizer state. This change would also allow us to add learning rate schedules.

optimize $\Lambda$, resulting in the following update:

$$(\Delta_\Lambda^{(t)}, \xi_\Lambda^{(t+1)}) = \text{Optimizer}(\overline{\Lambda}^{(t)}, \xi_\Lambda^{(t)}) \tag{3a}$$

$$\Lambda^{(t+1)} = \Lambda^{(t)} + \Delta_\Lambda^{(t)}. \tag{3b}$$

Because the above is adapting a neural network indirectly via another vector $\Lambda$ that is linearly mapped to the original parameter space, we refer to this as using a *linear adapter*, akin to the usage of "adapter" in the parameter-efficient finetuning literature ([Houlsby et al., 2019](#); [Hu et al., 2022](#)). Since $\mathbf{S}^\top \in \mathbb{R}^{d \times r}$, we have $\dim(\Delta_\Lambda^{(t)}) = \dim(\Delta_{\mathbf{S}\Theta}^{(t)})$ and $\dim(\xi_\Lambda^{(t)}) = \dim(\xi_{\mathbf{S}\Theta}^{(t)})$, i.e., the output and states of our optimizers have the exact same dimension in both cases. This is not a coincidence: we now show that optimizing this linear adapter when $\Lambda$ is initialized to $\mathbf{0}$ is *equivalent* to optimizing $\Theta$ in the original neural network with linearly transformed gradient dynamics. (For this and all subsequent proofs, refer to App. [A](#)).

**Theorem 1** (Equivalence of gradient transformations and linear adapters). *Suppose we are given initial parameters $\Theta^{(0)}$ and state $\xi_{\mathbf{S}\Theta}^{(0)}$. Let $\Theta^{(t)}$ be the parameters after $t$ update steps with the linearly transformed gradient dynamics with $\mathbf{S}$. Now consider a linear adapter which reparameterizes the model as $\Theta^{(0)} + \mathbf{S}^\top \Lambda$, where $\Lambda^{(0)}$ is initialized to $\mathbf{0}$ and the optimizer state $\xi_\Lambda^{(0)}$ is initialized to $\xi_{\mathbf{S}\Theta}^{(0)}$, and only $\Lambda$ is optimized. Then we have $\Theta^{(t)} = \Theta^{(0)} + \mathbf{S}^\top \Lambda^{(t)}$ for all $t$, i.e., the optimization trajectories are equivalent.*

*Remark.* The above only requires that the reparameterized model is equivalent to the original model at initialization, and can therefore be straightforwardly extended to cases where the adapter is not initialized to $\mathbf{0}$, as long as we have $\Theta^{(0)} = \tilde{\Theta} + \mathbf{S}^\top \Lambda^{(0)}$ for some $\tilde{\Theta}$ and $\Lambda^{(0)}$.

*Remark.* The above theorem holds for any optimizer of the form in eq. ([2](#)), e.g, Adam ([Kingma & Ba](#), [2015](#)). Notably, AdamW ([Loshchilov & Hutter](#), [2019](#)) does not fit this definition due to the way that weight decay is applied. See App. [B](#) for a discussion about weight decay, and what adjustments are required to preserve the equivalence.

### 3.2. Kronecker-factored Gradient Transformations

The formulation in Thm. [1](#) assumes very little about the neural network being trained and the gradient transformation (or, equivalently, linear adapter) being applied, which makes it difficult to enable practical memory savings. Concretely, consider applying an arbitrary linear transformation to just a single linear layer of a neural network with parameters $\mathbf{W} \in \mathbb{R}^{m \times n}$, i.e., $f(\mathbf{X}; \Theta) = \mathbf{W}\mathbf{X}$. In this case we have $\Theta = \text{vec}(\mathbf{W}) \in \mathbb{R}^{mn}$,[5] and thus arbitrary linear maps of the form $\mathbf{S} \in \mathbb{R}^{r \times mn}$ require $O(mnr)$ memory to store. This cost is already non-trivial for a single linear

---

[5] Intuitively, $\text{vec}(\cdot)$ sends a matrix to its vectorized form (i.e., stacks its columns into a vector), and $\text{vec}^{-1}(\cdot)$ is its inverse (i.e., unstacks vector back into matrix form).

layer of moderate size, and becomes rapidly intractable if we consider applying gradient transformations to the entirety of a model's parameters. As such, practical applications need to consider matrices $\mathbf{S}$ that are efficient to store in memory (and also efficient to apply to $\Theta$).

To this end, we consider *Kronecker-factored* linear maps of the form $\mathbf{S} = \mathbf{R}^\top \otimes \mathbf{L}$ where $\mathbf{L} \in \mathbb{R}^{d_L \times m}, \mathbf{R} \in \mathbb{R}^{n \times d_R}, d_L d_R = r$. This particular parameterization of $\mathbf{S}$ reduces the memory requirement to $O(d_L m + n d_R)$ and FLOPs to $\min\{O(d_L m n + d_L n d_R), O(mn d_R + d_L m d_R)\}$ (since $\mathbf{S}\overline{\Theta} = \text{vec}(\mathbf{L}\overline{\mathbf{W}}\mathbf{R})$), which can be memory-efficient if $d_L, d_R$ are small enough. We now show applying Thm. [1](#) to such an $\mathbf{S}$ establishes an equivalence between training with gradients transformed by $\mathbf{L}\overline{\mathbf{W}}\mathbf{R}$, and reparameterizing the linear layer as $\mathbf{W} + \mathbf{L}^\top \mathbf{A} \mathbf{R}^\top$ and only training $\mathbf{A} \in \mathbb{R}^{d_L \times d_R}$.

**Proposition 2** (Kronecker-factored parameterization of the linear map). *Let $\mathbf{W} \in \mathbb{R}^{m \times n}$ be the parameter matrix of a linear layer with corresponding gradient matrix $\overline{\mathbf{W}} \in \mathbb{R}^{m \times n}$. Further let $\Theta = \text{vec}(\mathbf{W})$ and $\overline{\Theta} = \text{vec}(\overline{\mathbf{W}})$. Consider training $\Theta$ as above with $\mathbf{S} = \mathbf{R}^\top \otimes \mathbf{L}$, i.e., by transforming the gradient matrix via $\mathbf{L}\overline{\mathbf{W}}\mathbf{R}$. Then the optimizer trajectory of $\mathbf{W}$ is equivalent to reparameterizing the model as $\mathbf{W} = \mathbf{W}^{(0)} + \mathbf{L}^\top \mathbf{A} \mathbf{R}^\top$, and then just training $\mathbf{A}$ (after initializing $\mathbf{A}^{(0)} = \mathbf{0}$).*

*Remark.* Prop. [2](#) shows that MoRA ([Jiang et al., 2024](#)), LoRA-XS ([Bałazy et al., 2024](#)), and PMSS ([Wang et al., 2025](#)), which are recent approaches to parameter-efficient finetuning which reparameterize a linear layer as $\mathbf{W} + \mathbf{B}\mathbf{A}\mathbf{C}$ and only train $\mathbf{A}$, can be interpreted as training the model with linearly-transformed gradients where the linear transformation has a Kronecker factorization.

Finally, as a simple corollary we now show that one can set $\mathbf{S}$ in a way that recovers GaLore, which in reparameterized form corresponds to one-sided LoRA, i.e., fixing one of the adapter matrices and only the training the other.

**Corollary 3** (GaLore is one-sided LoRA). *Let $\mathbf{W} \in \mathbb{R}^{m \times n}$ be the parameter matrix of a linear layer with corresponding gradient matrix $\overline{\mathbf{W}} \in \mathbb{R}^{m \times n}$. Without loss of generality, assume $m \leq n$. Now consider training $\mathbf{W}$ with* Optimizer *using GaLore, i.e., where we linearly transform the gradient matrix with a matrix $\mathbf{P}$, apply the optimizer, and transform the update via $\mathbf{P}^\top$, viz.,*

$$(\Delta_{\mathbf{W}}^{(t)}, \xi_{\mathbf{W}}^{(t+1)}) = \text{Optimizer}(\text{vec}(\mathbf{P}\overline{\mathbf{W}}^{(t)}), \xi_{\mathbf{W}}^{(t)})$$

$$\mathbf{W}^{(t+1)} = \mathbf{W}^{(t)} + \mathbf{P}^\top \text{vec}^{-1}(\Delta_{\mathbf{W}}^{(t)})$$

*where $\mathbf{P}$ is an arbitrary matrix of size $\mathbb{R}^{d \times m}$ and $d \leq m$ controls the dimensionality of the transformation. Then the optimizer trajectory of this network is equivalent to a network trained with the reparameterization $\mathbf{W} = \mathbf{W}^{(0)} + \mathbf{P}^\top \mathbf{A}$, where only $\mathbf{A}$ is learned.*

| Method | Adapter Parameterization | Trained | Frozen | Persisted |
|---|---|---|---|---|
| Baseline | $\mathbf{W}$ | $\mathbf{W}$ | – | $\mathbf{W}$ |
| ReLoRA (Lialin et al., 2024) | $\mathbf{W} + \mathbf{BA}$ | $\mathbf{A}, \mathbf{B}$ | $\mathbf{W}$ | $\mathbf{W}, \mathbf{A}, \mathbf{B}$ |
| Gradient SVD (GaLore; Zhao et al., 2024) | $\mathbf{W} + \mathbf{P}^\top\mathbf{A}, \quad \mathbf{P}^\top = \mathrm{SVD}(\overline{\mathbf{W}})$ | $\mathbf{A}$ | $\mathbf{W}, \mathbf{P}$ | $\mathbf{W}, \mathbf{P}, \mathbf{A}$ |
| Gaussian (Flora; Hao et al., 2024) | $\mathbf{W} + \mathbf{P}^\top\mathbf{A}, \quad \mathbf{P} \sim k\mathcal{N}(\mathbf{0}, \mathbf{I})$ | $\mathbf{A}$ | $\mathbf{W}, \mathbf{P}$ | $\mathbf{W}, \mathbf{A}$ |
| Rademacher | $\mathbf{W} + \mathbf{P}^\top\mathbf{A}, \quad \mathbf{P} \sim k\,\mathrm{Unif}(\{-1, 1\})$ | $\mathbf{A}$ | $\mathbf{W}, \mathbf{P}$ | $\mathbf{W}, \mathbf{A}$ |
| Random Semi-orthogonal | $\mathbf{W} + \mathbf{P}^\top\mathbf{A}, \quad \mathbf{P}^\top\mathbf{P} = k\mathbf{I}$ | $\mathbf{A}$ | $\mathbf{W}, \mathbf{P}$ | $\mathbf{W}, \mathbf{P}, \mathbf{A}$ |
| Two-sided Gaussian | $\mathbf{W} + \mathbf{L}^\top\mathbf{A}\mathbf{R}^\top, \quad \mathbf{L}, \mathbf{R} \sim k\mathcal{N}(\mathbf{0}, \mathbf{I})$ | $\mathbf{A}$ | $\mathbf{W}, \mathbf{L}, \mathbf{R}$ | $\mathbf{W}, \mathbf{A}$ |
| Two-sided Gradient SVD | $\mathbf{W} + \mathbf{L}^\top\mathbf{A}\mathbf{R}^\top, \mathbf{L}^\top, \quad \mathbf{R}^\top = \mathrm{SVD}(\overline{\mathbf{W}})$ | $\mathbf{A}$ | $\mathbf{W}, \mathbf{L}, \mathbf{R}$ | $\mathbf{W}, \mathbf{L}, \mathbf{R}, \mathbf{A}$ |

**Table 1:** A summary of methods tested for our pretraining experiments, where we list the gradient transformation method (which is not relevant for Baseline/ReLoRA) and the corresponding adapter parameterization. We also break down the reparameterized model into trained and frozen components, alongside the the set of components that need to be persisted in memory; for methods that make use of easy-to-materialize random sketching matrices (e.g., Gaussian) one only needs to persist the random number generator *seeds* for the gradient transformation, saving memory. Random semi-orthogonal matrices—a tall/wide matrix whose columns/rows are orthonormal vectors—are also random but are not straightforwardly materializable from a seed, and hence may need to be persisted across optimization steps. In the Gaussian and Rademacher cases, we use $k$ as shorthand for the constant that ensures that $\mathbb{E}[\mathbf{PP}^\top] = \mathbf{I}$.

*Remark.* The original GaLore work advocates for swapping out the gradient transformation every 200 optimizer steps. This does not break the equivalence in Cor. 3. In the adapter formulation, recomputing the gradient transformation corresponds to merging the learned adapter into the frozen weights, updating the frozen part of the adapter, and resetting the learned part to zero. This effectively amounts to ReLoRA (Lialin et al., 2024), where one side of the adapter is kept frozen throughout training.

While Prop. 2 and Cor. 3 focus on the case of a single linear layer, it is straightforward to generalize them to multiple linear layers. For example, one could treat the parameters of all layers as a single vector living in the product space of the individual layers' parameter spaces, and define the gradient transformation map $\mathbf{S}$ on that space as applying the correct projection to each of the layers' parameters individually. This can be implemented by modifying the optimizer step to apply a separate linear transformation to each layer.

Finally, we note that Hao et al. (2024) and Loeschcke et al. (2024) also show that training LoRA adapters with one side frozen with ordinary SGD is equivalent to applying a linear transformation to the gradient matrix, and Muhamed et al. (2024) further shows this for more general optimizers as in Cor. 3. Our Thm. 1 can be thus seen as a generalization of these recent results, where we show that this equivalence generalizes to arbitrary parameters of the neural network.

## 4. Empirical Study

The equivalences in §3 are agnostic to the choice of left and right transformations in $\mathbf{S} = \mathbf{R}^\top \otimes \mathbf{L}$. However, one might expect that the choice of $\mathbf{L}$ and $\mathbf{R}$ should matter for downstream performance. Hence, in the following sections, we first explore how the choice of $\mathbf{S}$ affects pretraining[6] per-

formance, and how by viewing gradient transformations as adapters, we further improve memory efficiency by combining the technique with QLoRA-style (Dettmers et al., 2023) training (§4.1). We then show how the converse is also useful: by viewing LoRA adapters through the lens of gradient transformations, we can improve distributed training of LoRA adapters by coordinating the LoRA adapter initialization across different workers (§4.2).

**Experimental setup.** We consider two moderate-scale language modeling settings: a 200M setting (training on 5B tokens) and a 1.3B setting (training on 10B tokens).[7] We use the Llama Transformer architecture (Touvron et al., 2023a) and train on the SlimPajama (Soboleva et al., 2023) dataset, tokenized using the Llama-2 (Touvron et al., 2023b) tokenizer, using sequences of length 2048. All numbers we report are perplexity on a disjoint (validation) set of SlimPajama. We use AdamW (Loshchilov & Hutter, 2019) with weight decay 0.1, $\beta_1 = 0.9$ and $\beta_2 = 0.95$. We warm up the learning rate to $4 \times 10^{-4}$, before decaying it via a cosine decay schedule to $1 \times 10^{-4}$. We conduct all training in `bfloat16` precision. See App. D for more details on our experimental setup.

### 4.1. Study 1: Memory-Efficient Pretraining

The discussion in §3 establishes a direct link between GaLore and one-sided LoRA. But how should we set $\mathbf{S}$ in practice? From the perspective of accurate gradient estimation, it would be ideal to have $\mathbf{S}^\top\mathbf{S} \approx \mathbf{I}$, since in the vanilla SGD case this would be equivalent to performing SGD with *sketched gradients*, where $\mathbf{S}^\top\mathbf{S}\overline{\Theta} \approx \overline{\Theta}$ (Murray et al., 2023). For the GaLore case with $\mathbf{S} = \mathbf{I} \otimes \mathbf{P}$, this amounts to setting $\mathbf{P}$ such that $\mathbf{PP}^\top \approx \mathbf{I}$, which could be achieved by, e.g., using random sketching matrices with the

---

[6]We target the pretraining setting as the gap between ordinary training and memory-efficient training methods is typically larger in pretraining than it is in finetuning.

[7]While this is not large by modern standards, due to our limited compute resources this is the largest setting at which we can feasibly perform experiments.

| Model | 200M | | 1.3B | |
|---|---|---|---|---|
| | PPL | Mem. | PPL | Mem. |
| Full pretraining | 18.58 | 1.32 | 12.44 | 8.04 |
| ReLoRA | 20.40 | 1.03 | 13.94 | 5.77 |
| QGaLore (`INT8`) | 23.86 | 0.94 | 15.23 | 5.15 |
| Gradient SVD (GaLore) | 21.34 | 0.96 | 13.62 | 5.27 |
| + `INT8` | 21.38 | 0.81 | 13.65 | 4.06 |
| + `NF4` (LoQT) | 26.52 | 0.73 | 16.10 | 3.46 |
| Gaussian (Flora) | 20.57 | 0.93 | 13.88 | 5.02 |
| + `INT8` | 20.55 | 0.78 | 13.87 | 3.81 |
| + `NF4` | 23.61 | 0.70 | 15.64 | 3.21 |
| Rademacher | 20.24 | 0.93 | 13.86 | 5.02 |
| + `INT8` | 20.26 | 0.78 | 13.78 | 3.81 |
| + `NF4` | 23.37 | 0.70 | 15.64 | 3.21 |
| Random Semi-orthogonal | 20.13 | 0.96 | 13.71 | 5.27 |
| + `INT8` | 20.32 | 0.81 | 13.75 | 4.06 |
| + `NF4` | 23.41 | 0.73 | 15.44 | 3.46 |
| Two-sided Gaussian | 23.98 | 0.93 | 15.28 | 5.02 |
| + `INT8` | 23.94 | 0.78 | 15.20 | 3.81 |
| + `NF4` | 27.93 | 0.70 | 16.95 | 3.20 |
| Two-sided Gradient SVD | 22.26 | 1.13 | 14.27 | 6.55 |
| + `INT8` | 22.08 | 0.97 | 14.16 | 5.35 |
| + `NF4` | 26.81 | 0.90 | 17.14 | 4.74 |

**Table 2:** Pretraining results at 200M and 1.3B scales. We report validation perplexity and estimated memory requirements (excluding activations) in GBs. GaLore + `NF4` quantization is equivalent to LoQT (Loeschcke et al., 2024). QGaLore (Zhang et al., 2024) quantizes both the base weights and the SVD projection to `INT8`, but does not adopt the LoRA parameterization.

property $\mathbb{E}[\mathbf{P}\mathbf{P}^\top] = \mathbf{I}$.[8] As noted by Hao et al. (2024), using a random sketching matrix can enable further savings as only the random number generator (RNG) seed needs to be persisted across optimization steps. We thus experiment with a variety of sketching matrices for LoRA-based pretraining as shown in Tab. 1.

Another benefit of the adapter parameterization of gradient projections is that it allows us to be more memory efficient by quantizing the base weights as done in QLoRA (Dettmers et al., 2023). Specifically, given the adapter parameterization $\Theta + \mathbf{S}^\top \Lambda$ we can quantize $\Theta$ and only train $\Lambda$, thus enabling further memory savings. Finally, the adapter parameterization has the additional benefit of reducing the number of trainable parameters being registered for automatic differentiation, which allows for gradient accumulation to happen in a lower dimensional space. (See App. D for more discussion.)

**Results.** The results are shown in Tab. 2, where we follow the original GaLore paper and use a rank of 256 for the 200M model and a rank of 512 for the 1.3B model,[9] and

[8]This sketching view of LoRA provides a possible perspective on why one-sided LoRA finetuning works well in practice (Zhang et al., 2023; Zhu et al., 2024; Hayou et al., 2024).

[9]The only exceptions are for the double-sided methods. For the two-sided Gaussian, we set the rank as to match the number of

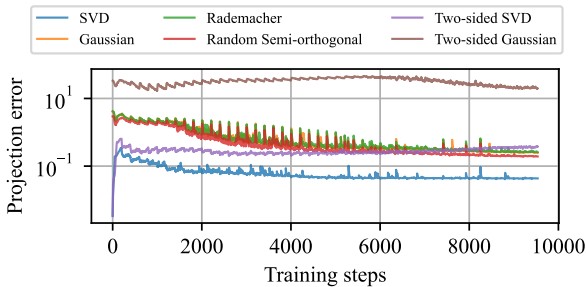

**Figure 1:** Average of gradient reconstruction error $\|\overline{\Theta} - \mathbf{S}^\top \mathbf{S}\overline{\Theta}\|_2^2$ of the various transformations across training steps at 200M scale.

further merge the adapters into the full weights and reinitialize them every 200 steps. We see that one-sided transformations, regardless of their nature, perform somewhat similarly at both 200M and 1.3B scale, suggesting that using a random gradient transformation matrix that can be cheaply rematerialized on-the-fly may be more economical than using the top singular vectors derived from the gradient as in GaLore. We also find that ReLoRA performs comparably to one-sided gradient transformations, suggesting that the additional flexibility of ReLoRA (i.e., optimizing two sides of a LoRA adapter instead of only one side) is not necessary. Using two-sided Gaussian gradient transformations degrades performance when memory consumption is matched to one-sided methods; two-sided SVD-based projections fare slightly better but still trail behind one-sided methods and incur a much larger memory cost, since two projection matrices must be persisted. While Zhao et al. (2024) report no gap between GaLore and full pretraining, we did not find this to be true on our setup,[10] and instead observe a non-trivial gap between regular (full) training and these memory-efficient pretraining methods.

When adding quantization to the base weights (where we use groups of size 256), we find that, across the board, 8-bit integer quantization can be performed without major performance degradation, whereas 4-bit NormalFloat quantization begins to incur a penalty (4-bit integer did even worse). This suggests that adapter-based training, with a rematerializable gradient transformation and 8-bit integer quantization of the base weights, is a promising recipe for memory-efficient pretraining. Finally, we find that QGaLore (Zhang et al., 2024), which quantizes the weights to `INT8` precision and trains these `INT8` weights directly using an SVD gradient transformation, underperforms a QLoRA-style approach to quantized GaLore training.

trainable parameters in the one-sided Gaussian approach. For the two-sided SVD, we use the same rank as in two-sided Gaussian, which incurs more memory since the projection matrices must be persisted across optimization steps.

[10]Which is different from theirs in many ways, e.g., we train on longer sequences with gradient accumulation using `bfloat16`.

| Method | Projection Init. | 200M | 1B |
|---|---|---|---|
| Dist. Training (DiLoCo) | – | 18.00 | 12.77 |
| Dist. ReLoRA (LTE) | – | 20.97 | 13.72 |
| Identical Random | $\mathbf{P}_i = \mathbf{P}_j$ | 21.51 | 14.28 |
| Independent Random | $\mathbb{E}[\mathbf{P}_i \mathbf{P}_j^\top] = \mathbf{0}$ | 20.11 | 13.66 |
| Distributed Random | $\mathbf{P}_i \mathbf{P}_j^\top = \mathbf{0}$ | 19.81 | 13.51 |

**Table 3:** Results of the distributed training experiments, where four workers are trained independently and synchronized every 500 steps, following DiLoCo (Douillard et al., 2024). We use random semi-orthogonal matrices for the distributed (one-sided) LoRA experiments. For the (re)initializations of worker-specific projections $\{\mathbf{P}_k\}_{k=1}^K$, *identical* shares the projection matrix across workers, *independent* initializes each worker's projection independently, and *distributed* initializes the worker projections such that they are all orthogonal to one another. The top two rows are our baselines, viz., DiLoCo and a distributed variant of ReLoRA, which is similar to LTE (Huh et al., 2024).

**Analysis.** We have motivated our experiments with sketching matrices from the perspective of accurate gradient compression, i.e., we use $\mathbf{S}$ to compress the gradient, and then $\mathbf{S}^\top$ to decompress it. From this compression viewpoint, one may then wonder whether different gradient transformations exhibit different reconstruction capabilities, and whether this ultimately dictates the performance of the resulting model. As shown in Fig. 1, we find that the gradient reconstruction error does not correlate with performance. As expected, methods that perform SVD on the gradients have low reconstruction error (since SVD explicitly minimizes a reconstruction objective), but as shown in Tab. 2, SVD performs similarly to sketching matrices, which have higher reconstruction error. We believe that the relationship between the nature of the gradient transformation and downstream performance is fairly complex, and merits further investigation. Fig. 2 of App. C shows similar results for cosine similarity instead of squared error.

### 4.2. Study 2: Distributed Pretraining

Our second experiment targets distributed pretraining of LLMs across poorly-connected and resource-constrained workers, which is important for many applications of interest, from federated training of LLMs to scaling up LLMs across data centers that are not co-located, i.e., where techniques like FSDP are not possible. DiLoCo (Douillard et al., 2024) is a recent and effective approach that has workers train independently for some number of iterations using an inner optimizer, and then uses the average change in parameters from each worker as a "pseudo-gradient" on an outer optimizer that updates a global copy of the parameters (i.e., as in federated learning; McMahan et al., 2017; Reddi et al., 2021). This approach has since been scaled up to train 10B LLMs across distributed workers.[11]

However, DiLoCo still assumes that each worker has enough memory to perform a full forward/backward pass on the model, i.e., it does not target memory efficiency. A memory-efficient distributed training approach that is of particular interest in light of the equivalence in §3 is LoRA-the-explorer (LTE; Huh et al., 2024), which can be seen as an extension of ReLoRA to the distributed setting. LTE has $K$ independent workers train separate LoRA adapters for a small number of local steps, and then performs a global step by averaging the adapters across workers. The globally-averaged adapter is then merged into the base weights, and optimization continues by resetting and traiing the worker-specific LoRA adapters.

We will now describe how the equivalence in §3 can be used to derive an improved version of LTE, which trains only one side of the LoRA adapter in each worker, but initializes the frozen side in a worker-aware manner. Consider a one-sided analogue of LTE, where the weight $\mathbf{W}_k^{(g,l)}$ for the $k$th worker after $g$ global and $l$ local updates is

$$\mathbf{W}_k^{(g,l)} = \mathbf{W}_k^{(g,0)} + \mathbf{P}_k^{(g)^\top} \mathbf{A}_k^{(g,l)}$$

and only $\mathbf{A}$ is trained. The global step is given by,

$$\mathbf{W}_k^{(g+1,0)} = \mathbf{W}_k^{(g,0)} + \frac{1}{K} \sum_{k=1}^K \mathbf{P}_k^{(g)^\top} \mathbf{A}_k^{(g,L)}$$

where we have assumed that the global step is performed after $L$ local steps. After a global step, we would also reset $\mathbf{A}$ by setting $\mathbf{A}_k^{(g+1,0)} = \mathbf{0}$ and similarly swap out $\mathbf{P}_k$ for another (e.g., random) matrix for all $k$.

By Cor. 3, local steps must correspond to training the worker weights using a gradient transformation,

$$\mathbf{W}_k^{(g,l)} = \mathbf{W}_k^{(g,0)} + \mathbf{P}_k^{(g)^\top} \Delta_{\mathbf{PW}}^{(g,l)},$$

where we use $\Delta_{\mathbf{PW}}^{(g,l)}$ to denote the optimizer update that was performed in the lower dimensional transformed space. Further, a global step in this view can be equivalently seen as defining a global pseudo-gradient $\Delta^{(g)}$ as the average of for the local pseudo-gradients $\{\Delta_1^{(g)}, \ldots, \Delta_K^{(g)}\}$,

$$\Delta^{(g)} = \frac{1}{K} \sum_{k=1}^K \Delta_k^{(g)}, \quad \Delta_k^{(g)} = \mathbf{W}_k^{(g,L)} - \mathbf{W}_k^{(g,0)}$$

The global weight update is then given by a step using the global pseudo-gradient,

$$\mathbf{W}_k^{(g+1,0)} = \mathbf{W}^{(g,0)} + \Delta^{(g)}.$$

This can be straightforwardly generalized to the use of different learning rates and more advanced optimizers.

One approach to initialize/reset the frozen side of worker-specific LoRA adapters (i.e., the gradient projections $\mathbf{P}_k$) is to sample a projection and broadcast it to all workers. However, the GaLore–LoRA duality suggests a different scheme. Thm. 1 shows that training with linear gradi-

| Method | (Rank, Workers) | | |
|---|---|---|---|
| | (128, 8) | (256, 4) | (512, 2) |
| Dist. Training (DiLoCo) | 17.81 | 18.00 | 18.56 |
| Dist. ReLoRA (LTE) | 23.76 | 20.97 | 19.54 |
| Identical Random | 23.96 | 21.51 | 20.32 |
| Independent Random | 20.64 | 20.11 | 19.97 |
| Distributed Random | 20.32 | 19.81 | 19.66 |

**Table 4:** Results of the distributed pretraining experiments as we vary the rank of the gradient projections and number of workers. For the DiLoCo baseline, we only vary the number of workers, which means that as we increase the number of workers, the DiLoCo baseline can only benefit since we are training on more data without any downside (i.e., a rank restriction). Note that the for the distributed ReLoRA baseline (which is similar to LTE, Huh et al., 2024), we have double the number of trainable parameters as in the one-sided methods.

ent transformations only optimizes a *subspace* of the full model, namely $\text{range}(\mathbf{S}^\top)$ where $\mathbf{S}^\top = \mathbf{I} \otimes \mathbf{P}^\top$ in the GaLore case. This suggests a different approach to distributed LoRA training, wherein the frozen part of each LoRA adapter, $\mathbf{S}_1, \ldots, \mathbf{S}_K$, is initialized differently, so that the sum of their ranges allows a larger subspace to be trained. For example, we could sample random semi-orthogonal matrices $\mathbf{P}_1, \ldots, \mathbf{P}_K$ uniformly at random, assign each worker $\mathbf{S}_i = \mathbf{I} \otimes \mathbf{P}_i^\top$, and they will likely each cover different portions of the space. An even stronger strategy would be to demand that $\text{range}(\mathbf{S}_1^\top), \ldots, \text{range}(\mathbf{S}_K^\top)$ must be mutually orthogonal, which can be realized by keeping the $\mathbf{P}_i$'s as semi-orthogonal, but enforcing that $\mathbf{P}_i \mathbf{P}_j^\top = \mathbf{0}$ (e.g., by generating a random $m \times m$ orthogonal matrix and having each worker take a different $d \times m$ submatrix.) Intuitively, this ensures that no worker is duplicating the work of another, since their projections are pairwise orthogonal. We experiment with such *identical random*, *independent random*, *distributed random* initialization schemes.

**Results.** The results for the main set of experiments are shown in Tab. 3. We consider two baselines: (i) DiLoCo (Douillard et al., 2024), which has each worker training independently for 500 steps before computing a pseudo-gradient that is used to update the global parameters using SGD with Nesterov momentum (Sutskever et al., 2013), and (ii) distributed ReLoRA, which is an analog of ReLoRA but adapted to train like DiLoCo, i.e., one trains the LoRA adapter for 500 steps and defines the adapter weight as the pseudo-gradient for the Nesterov step; this is very close to LTE.[12] Our distributed GaLore experiments make use of random semi-orthogonal projections since the distributed random initialization for it is easy to compute, and does not add significant communication overhead.[13]

As in §4.1, distributed GaLore leads to degradations at both 200M and 1.3B scales compared to the full distributed training baseline (i.e., DiLoCo). However, our distributed random initialization scheme, where workers are "aware" of each, performs well, thus demonstrating the utility of the gradient transformation–adapter duality from §3.

**Analysis.** We perform a study at the 200M scale over how the number of workers and rank affect performance. Intuitively, larger ranks lead to a larger subspace being trained by each worker (and, in the limit, we should recover something akin to DiLoCo when there is no rank reduction), so we would expect performance to improve as we increase the rank. Indeed, the results for this ablation (shown in Tab. 4) confirm this intuition, likely because DiLoCo benefits from more workers to get a better estimate of the pseudo-gradient for the outer optimizer step. More surprisingly, we find that this gap is largely bridged by ensuring that different gradient transformations are assigned to each worker, with the distributed initialization once again performing the best. It would be interesting to further study how the effectiveness of the distributed initialization scheme changes as we go to more extreme settings (e.g., hundreds of extremely low-rank workers).

## 5. Discussion and Limitations

The preceding studies focus on two situations in which the duality between linear adapters and gradient transformations offers practical insights. We believe there are many other avenues that merit further exploration. For instance, Thm. 1 makes no assumptions about the structure of $\mathbf{S}$; while we only considered Kronecker-factorized matrices, other linear maps that admit efficient storage and computation would be interesting to explore. The transformation also just needs to be applied to a gradient vector, suggesting that one could compress the gradient of multiple layers in a network jointly. This amounts to sharing LoRA parameters across layers on the GaLore–LoRA case, which has been found to be successful for parameter-efficient fine-tuning (Renduchintala et al., 2024; Song et al., 2024). eegardless of the structure of $\mathbf{S}$, as discussed in §4.1, what characterizes a good $\mathbf{S}$ is not clear but has a large impact. It may be possible to *learn* a good $\mathbf{S}$ with meta-learning-style approaches, which can be seen as *learning an optimizer* (Andrychowicz et al., 2016; Li & Malik, 2016; Wichrowska et al., 2017; Bello et al., 2017, *i.a.*).[14] Finally, while we focused on linear gradient transformations, where we proved exact equivalence with a linear adapter parameterization, it may be possible to establish approxi-

---

[12]LTE can be seen as using SGD as the optimizer on the pseudo-gradients, but we found this led to worse results in preliminary experiments.

[13]Each worker just needs the seed used to sample the orthogonal matrix, and the indices of the rows it will keep.

[14]In the GaLore/LoRA case, learning $\mathbf{P}$ in this meta-learning sense is different from learning $\mathbf{P}$ in the ordinary LoRA sense, i.e., when both $\mathbf{P}$ and $\mathbf{A}$ are trained with gradient descent against the same loss function.

mate equivalence between non-linear gradient transformations and other types of adapters.

Our work has several limitations. Due to compute constraints, we were only able to scale our experiments to 1.3B, which is small by industry standards. While our duality results are more general, our experiments primarily focus on the special case of the GaLore–LoRA duality. We chose to focus primarily on a wide array of gradient transformations, but forgo a study of the interaction between such transformations and the choice of optimizer, projection reinitialization schedule, etc. Ultimately, we believe that our results signal that these techniques could be applied at larger scales, especially when performing distributed training in memory-constrained regimes.

## 6. Related Work

**Memory-efficient training.** There is a growing body of research focused on memory-efficient LLM training. This work explores the connections among GaLore (Zhao et al., 2024), LoRA (Hu et al., 2022), QLoRA (Dettmers et al., 2023), and ReLoRA (Lialin et al., 2024). Various approaches in low-rank adaptations have been proposed to enhance these techniques (Renduchintala et al., 2024; Sheng et al., 2024; Zhang et al., 2023; Xia et al., 2024; Wang et al., 2023b; Hao et al., 2024; Wang et al., 2025), including efforts to train models from scratch (Kamalakara et al., 2022; Wang et al., 2023a; Zhao et al., 2023). Broadly, memory-efficient training also encompasses methods such as adapters (Houlsby et al., 2019; Karimi Mahabadi et al., 2021), which insert trainable layers and prompt tuning (Li & Liang, 2021; Lester et al., 2021), which optimizes continuous prompts. Additionally, its combination with quantization techniques (Kwon et al., 2022) and other methods that update subparts of the parameter vector (Guo et al., 2021; Ben Zaken et al., 2022; Sung et al., 2021) are also relevant.

**Memory-reduction via randomization.** Randomization has been used in other contexts to reduce memory consumption in automatic differentiation. Adelman et al. (2021) and Liu et al. (2023) perform row/column subsampling to reduce the amount of computation and memory required to compute gradients. Bershatsky et al. (2022) also explores Gaussian projections, but in the context of reducing activation memory by sketching them. Oktay et al. (2021) construct gradient estimators by computing the gradient on a subsample of the paths in the computation graph. More tangentially, MeZO (Malladi et al., 2023) amounts to sketching the gradient of a neural network by performing forward-mode automatic differentiation on random vectors.

## 7. Conclusion

We proved a general equivalence between training an LLM with linear transformations of gradients and training with additive linear adapters, and showed the GaLore–LoRA equivalence is a special case of this result. We then used this equivalence to derive more memory-efficient and performant methods for LLM pretraining, including combinations of quantization and gradient-projection methods and improved initialization for distributed adapter pretraining.

## Impact Statement

The last few years have seen widespread interest in LLMs. Perhaps the most salient finding from the race to build the best LLMs is that increasing parameter counts in tandem with data is of paramount importance. This makes it very hard to train competitive LLMs unless one has the best and latest hardware, which offers the most memory capacity and thus the ability to actually train these models in practice. Our research targets exactly this setting, offering a mathematical connection between two methods at the cornerstone of memory-efficient training, and showing how this connection can lead to further improvements in memory-efficiency and distributed training.

## Acknowledgements

We thank Shannon Zejiang Shen, Li Du, Aniruddha Nrusimha, Jeremy Bernstein, Jyothish Pari, Sami Jaghouar, Johannes Hagemann, and the anonymous reviewers for helpful discussions and feedback. This work was partially supported by the National Science Foundation under CAREER Award No. 2441872 and the MIT-IBM Watson AI Lab.

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

## A. Proofs

### A.1. Proof of Thm. 1

**Theorem 1** (Equivalence of gradient transformations and linear adapters)**.** *Suppose we are given initial parameters $\Theta^{(0)}$ and state $\xi_{\mathbf{S}\Theta}^{(0)}$. Let $\Theta^{(t)}$ be the parameters after $t$ update steps with the linearly transformed gradient dynamics with $\mathbf{S}$. Now consider a linear adapter which reparameterizes the model as $\Theta^{(0)} + \mathbf{S}^\top \Lambda$, where $\Lambda^{(0)}$ is initialized to $\mathbf{0}$ and the optimizer state $\xi_\Lambda^{(0)}$ is initialized to $\xi_{\mathbf{S}\Theta}^{(0)}$, and only $\Lambda$ is optimized. Then we have $\Theta^{(t)} = \Theta^{(0)} + \mathbf{S}^\top \Lambda^{(t)}$ for all $t$, i.e., the optimization trajectories are equivalent.*

*Proof.* To show that the two optimization trajectories are equivalent, we will use induction to show that after every optimizer step $t \geq 0$ we have that the optimizer states are equivalent, i.e., $\xi_\Lambda^{(t)} = \xi_{\mathbf{S}\Theta}^{(t)}$, which in turn allows us to show that the networks are identical, i.e., $\Theta^{(t)} = \Theta^{(0)} + \mathbf{S}^\top \Lambda^{(t)}$.

Note that at initialization, since $\Lambda^{(0)} = \mathbf{0}$, we have that

$$\Theta^{(0)} + \mathbf{S}^\top \Lambda^{(0)} = \Theta^{(0)} + \mathbf{S}^\top \mathbf{0} = \Theta^{(0)},$$

which implies that our reparameterized network is identical to our original network. By assumption we also have that the optimizer states are equal $\xi_\Lambda^{(0)} = \xi_{\mathbf{S}\Theta}^{(0)}$. Now assume that this is true for $t \leq k$, i.e., for all $t \leq k$

$$\Theta^{(0)} + \mathbf{S}^\top \Lambda^{(t)} = \Theta^{(t)} \qquad \text{(Neural networks equivalent)} \tag{4}$$

$$\xi_\Lambda^{(t)} = \xi_{\mathbf{S}\Theta}^{(t)}. \qquad \text{(Optimizer states equivalent)} \tag{5}$$

Now note that for $k+1$,

$$\Theta^{(0)} + \mathbf{S}^\top \Lambda^{(k+1)} = \Theta^{(0)} + \mathbf{S}^\top (\Lambda^{(k)} + \Delta_\Lambda^{(k)}) = \Theta^{(k)} + \mathbf{S}^\top \Delta_\Lambda^{(k)} \tag{6}$$

where we used eq. (3) in the first equality, and eq. (4) for the second equality. Also by eq. (4) and by the chain rule, we have that the gradients of the loss function at timestep $t$ to the (effective) parameters of the networks are the same, i.e., $\overline{\Theta^{(0)} + \mathbf{S}^\top \Lambda^{(k)}} = \overline{\Theta^{(k)}}$. In particular, this means that

$$\overline{\Lambda^{(k)}} = \mathbf{S}\overline{\mathbf{S}^\top \Lambda^{(k)}} = \mathbf{S}\overline{\Theta^{(0)} + \mathbf{S}^\top \Lambda^{(k)}} = \mathbf{S}\overline{\Theta^{(k)}} \tag{7}$$

where the first two equalities follow from the chain rule. Specifically, the first equality follows from writing the derivative of the loss w.r.t. the adapter parameter $\Lambda^{(k)}$ as a function of the derivative of the loss w.r.t. the additive component of the adapter, $\mathbf{S}^\top \Lambda^{(k)}$; the second equality follows from recognizing that the latter is equal to the derivative of the loss w.r.t. the effective value $\Theta^{(0)} + \mathbf{S}^\top \Lambda^{(k)}$ of the (adapted) parameter. In turn, expanding $\Delta_\Lambda^{(k)}$,

$$(\Delta_\Lambda^{(k)}, \xi_\Lambda^{(k+1)}) = \text{Optimizer}(\overline{\Lambda}^{(k)}, \xi_\Lambda^{(k)}) = \text{Optimizer}(\mathbf{S}\overline{\Theta^{(k)}}, \xi_{\mathbf{S}\Theta}^{(k)}) = (\Delta_{\mathbf{S}\Theta}^{(k)}, \xi_{\mathbf{S}\Theta}^{(k+1)}), \tag{8}$$

where the first equality is the optimizer we use to train the reparameterized model in eq. (3), in the second equality we use eq. (7) and eq. (5). But by eq. (6),

$$\Theta^{(0)} + \mathbf{S}^\top \Lambda^{(k+1)} = \Theta^{(k)} + \mathbf{S}^\top \Delta_\Lambda^{(k)} = \Theta^{(k)} + \mathbf{S}^\top \Delta_{\mathbf{S}\Theta}^{(k)} = \Theta^{(k+1)}. \tag{9}$$

Eq. (8) proves optimizer states are equivalent for optimizer step $k+1$, whereas eq. (9) establishes the networks are equivalent for optimizer step $k+1$, thus completing the proof. $\qquad\square$

### A.2. Proof of Prop. 2

**Proposition 2** (Kronecker-factored parameterization of the linear map)**.** *Let $\mathbf{W} \in \mathbb{R}^{m \times n}$ be the parameter matrix of a linear layer with corresponding gradient matrix $\overline{\mathbf{W}} \in \mathbb{R}^{m \times n}$. Further let $\Theta = \text{vec}\,(\mathbf{W})$ and $\overline{\Theta} = \text{vec}\,(\overline{\mathbf{W}})$. Consider*

*training $\Theta$ as above with $\mathbf{S} = \mathbf{R}^\top \otimes \mathbf{L}$, i.e., by transforming the gradient matrix via $\mathbf{L}\overline{\mathbf{W}}\mathbf{R}$. Then the optimizer trajectory of $\mathbf{W}$ is equivalent to reparameterizing the model as $\mathbf{W} = \mathbf{W}^{(0)} + \mathbf{L}^\top \mathbf{A} \mathbf{R}^\top$, and then just training $\mathbf{A}$ (after initializing $\mathbf{A}^{(0)} = \mathbf{0}$).*

*Proof.* From Thm. 1, we know that training a linear layers using the gradient transformation $\mathbf{S} = \mathbf{R}^\top \otimes \mathbf{L}$ corresponds to using the reparameterization:

$$\Theta = \Theta^{(0)} + (\mathbf{R}^\top \otimes \mathbf{L})^\top \Lambda$$

and training $\Lambda$ instead, using the same optimizer. Letting $\text{vec}(\mathbf{A}) = \Lambda$, we then have

$$\text{vec}(\mathbf{W}) = \text{vec}\left(\mathbf{W}^{(0)}\right) + (\mathbf{R} \otimes \mathbf{L}^\top) \text{vec}(\mathbf{A})$$
$$= \text{vec}\left(\mathbf{W}^{(0)}\right) + \text{vec}(\mathbf{L}^\top \mathbf{A} \mathbf{R}^\top)$$

where in the first equation we used the fact that $(\mathbf{M} \otimes \mathbf{N})^\top = \mathbf{M}^\top \otimes \mathbf{N}^\top$ and in the second equation we used $\text{vec}(\mathbf{M}\mathbf{N}\mathbf{O}) = (\mathbf{O}^\top \otimes \mathbf{M}) \text{vec}(\mathbf{N})$. Taking $\text{vec}^{-1}(\cdot)$ in both sides completes the proof. $\square$

### A.3. Proof of Cor. 3

We now state a more general version of Cor. 3, and then prove it.

**Corollary 4** (Galore is one-sided LoRA (General)). *Let $\mathbf{W} \in \mathbb{R}^{m \times n}$ be the parameter matrix of a linear layer with corresponding gradient matrix $\overline{\mathbf{W}} \in \mathbb{R}^{m \times n}$. Consider training $\mathbf{W}$ with* Optimizer *using GaLore, i.e., where we linearly transform the gradient matrix with a matrix $\mathbf{P}$,*

$$\widetilde{\overline{\mathbf{W}}} = \begin{cases} \mathbf{P}\overline{\mathbf{W}} & m \leq n \quad \text{(i.e., apply from the left)} \\ \overline{\mathbf{W}}\mathbf{P} & m > n \quad \text{(i.e., apply from the right)} \end{cases}$$

*and then apply our optimizer on it, before transforming our update back to parameter space via $\mathbf{P}^\top$, viz.,*

$$(\Delta_{\mathbf{W}}^{(t)}, \xi_{\mathbf{W}}^{(t+1)}) = \text{Optimizer}(\text{vec}\left(\widetilde{\overline{\mathbf{W}}}^{(t)}\right), \xi_{\mathbf{W}}^{(t)})$$

$$\mathbf{W}^{(t+1)} = \begin{cases} \mathbf{W}^{(t)} + \mathbf{P}^\top \text{vec}^{-1}\left(\Delta_{\mathbf{W}}^{(t)}\right) & m \leq n \\ \mathbf{W}^{(t)} + \text{vec}^{-1}\left(\Delta_{\mathbf{W}}^{(t)}\right) \mathbf{P}^\top & m > n \end{cases}$$

*where $\mathbf{P}$ is an arbitrary matrix of size $\mathbb{R}^{d \times m}$ (if $m \leq n$) or $\mathbb{R}^{n \times d}$ (otherwise) and $d \leq \min(m, n)$ controls the dimensionality of the transformation. Then the optimizer trajectory of this network is equivalent to a network trained with the reparameterization:*

$$\mathbf{W} = \begin{cases} \mathbf{W}^{(0)} + \mathbf{P}^\top \mathbf{A} & m \leq n \\ \mathbf{W}^{(0)} + \mathbf{A}\mathbf{P}^\top & m > n, \end{cases}$$

*i.e., adding LoRA adapters where one side is frozen to $\mathbf{P}^\top$ and only the other side, $\mathbf{A}$, is learned.*

*Proof.* Define

$$\mathbf{S} = \begin{cases} \mathbf{I}_n \otimes \mathbf{P} & m \leq n \\ \mathbf{P}^\top \otimes \mathbf{I}_m & m > n \end{cases}$$

where $\mathbf{I}_m$ is the $m \times m$ identity matrix, and similarly for $\mathbf{I}_n$. Then note that

$$\mathrm{vec}\left(\widetilde{\overline{\mathbf{W}}}\right) = \begin{cases} \mathrm{vec}\left(\mathbf{P}\overline{\mathbf{W}}\right) & m \leq n \\ \mathrm{vec}\left(\overline{\mathbf{W}}\mathbf{P}\right) & m > n \end{cases}$$
$$= \mathbf{S}\,\mathrm{vec}\left(\overline{\mathbf{W}}\right)$$

using $\mathrm{vec}\left(\mathbf{MNO}\right) = \left(\mathbf{O}^\top \otimes \mathbf{M}\right)\mathrm{vec}\left(\mathbf{N}\right)$ as before. Similarly, we have that $\mathrm{vec}\left(\mathbf{W}^{(t+1)}\right) = \mathrm{vec}\left(\mathbf{W}^{(t)}\right) + \mathbf{S}^\top \Delta_{\mathbf{W}}^{(t)}$. Hence, by Thm. 1, we have that training a network with GaLore is equivalent to introducing a parameter $\mathbf{A}$ and optimizing using the reparameterization $\mathrm{vec}\left(\mathbf{W}\right) = \mathrm{vec}\left(\mathbf{W}^{(0)}\right) + \mathbf{S}\,\mathrm{vec}\left(\mathbf{A}\right)$. Observe that this choice of $\mathbf{S}$ is a special case of Prop. 2 where $\mathbf{L} = \mathbf{P}$ and $\mathbf{R} = \mathbf{I}$ (if $m \leq n$) or $\mathbf{L} = \mathbf{I}$ and $\mathbf{R} = \mathbf{P}$ (if $m > n$). Thus, the reparameterization corresponds to LoRA with one of the two adapter matrices frozen to $\mathbf{P}$. $\qquad\square$

## B. Weight Decay

Cor. 3 establishes an equivalence between GaLore and LoRA when stateful optimizers are in play (i.e., those satisfying eq. (2)). While Adam (Kingma & Ba, 2015) can be straightforwardly recast as a stateful optimizer, it turns out that weight decay, as is traditionally implemented in, e.g., AdamW (Loshchilov & Hutter, 2019), breaks this symmetry as it does not fit our definition of a stateful optimizer. From the perspective of our definition, the problem is that optimizer steps with AdamW are not solely a function of the observed gradients up until this point, but also the actual values of the parameters. This is important, since automatic differentiation libraries traditionally distinguish *trainable* and *non-trainable* parameters, with weight decay being applied to the former. Since the duality in Cor. 3 changes what the trainable parameters are, this means that the weight decay is applied differently in the gradient transformation view and in the linear adapter view.

For example, consider taking a linear layer with weight $\Theta$, and training it with an optimizer that applies weight decay. When training this layer with a linear gradient transformation $\mathbf{S}$, after a single optimizer step, our new weight is given by

$$(\Delta_\Theta^{(0)}, \xi_\Theta^{(1)}) = \mathrm{OptimizerWithoutWeightDecay}(\mathbf{S}\overline{\Theta}^{(0)}, \xi_\Theta^{(0)}) \tag{10}$$
$$\Theta^{(1)} = \Theta^{(0)} + \mathbf{S}^\top \Delta_\Theta^{(0)} - \lambda\Theta^{(0)} \tag{11}$$

where $\lambda \in \mathbb{R}^+$ is our weight decay penalty. Similarly, following Thm. 1, our linear layer's effective weight after a single optimizer step, in the adapter view, is given by

$$\Theta_{\mathrm{effective}}^{(1)} = \Theta^{(0)} + \mathbf{S}^\top(\Lambda^{(0)} + \Delta_\Lambda^{(0)} - \lambda\Lambda^{(0)}) \tag{12}$$
$$= \Theta^{(0)} + \mathbf{S}^\top(\Lambda^{(0)} + \Delta_\Lambda^{(0)}) - \lambda\mathbf{S}^\top\Lambda^{(0)} \tag{13}$$
$$= \Theta^{(0)} + \mathbf{S}^\top\Delta_\Theta^{(0)} - \lambda\mathbf{S}^\top\Lambda^{(0)} \tag{14}$$

where the final equality follows from Thm. 1. Note that in general, we do not have that $\Theta^{(0)} = \mathbf{S}^\top\Lambda^{(0)}$, which means the optimizer trajectories may diverge.

An alternative interpretation for the above is that weight decay can be seen (roughly) as placing a Gaussian prior on the trainable parameters, but since the set of trainable parameters is different under each view, the equivalence does not immediately hold. We note that it is not outright clear if one implementation of weight decay is superior, so further research is required in this regard. In our experiments, for simplicity, we leave the application of weight decay untouched, i.e., we (implicitly) use the implementation of weight decay that naturally arises from the adapter or gradient transformation views.

**Maintaining the equivalence.** If one truly cares about preserving the optimizer trajectory even when training with weight decay, practically all one has to do is adjust the application of the weight decay so that it reflects the behavior of weight decay in the gradient transformation view or in the linear adapter view. Adjusting the linear adapter weight decay application to match the gradient transformation weight decay application is fairly simple: one just has to compute the effective weight at every timestep, as done above, and decay the frozen base weights directly. The converse is possible but slightly trickier, since the application of weight decay in the linear adapter view requires one to know what $\Lambda^{(t)}$ is, which may require the

introduction of additional optimizer state. For example, one could store $\Theta^{(0)}$ and solve[15] $\mathbf{S}^\top \Lambda^{(t)}_{\text{effective}} = \Theta^{(t)} - \Theta^{(0)}$ on every optimizer state,[16] or one could store and continually update $\Lambda_{\text{effective}}$ as part of the optimizer state.

## C. Additional Results

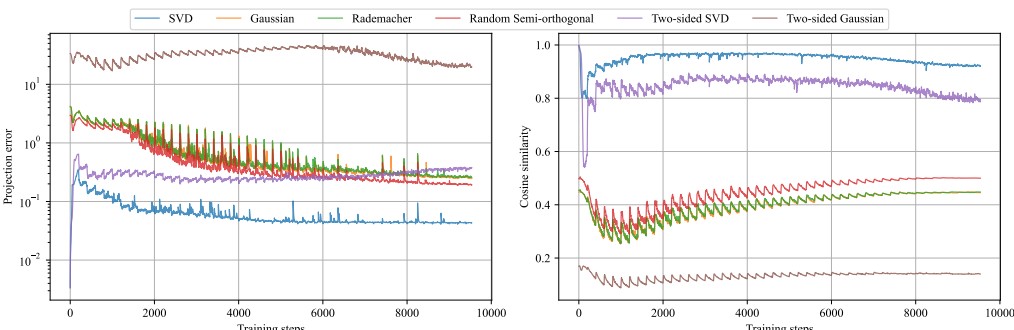

**Figure 2:** Full results for gradient reconstruction error (i.e., $\|\overline{\Theta} - \mathbf{S}^\top \mathbf{S} \overline{\Theta}\|^2$) (left) and cosine similarity (i.e., $\cos(\overline{\Theta}, \mathbf{S}^\top \mathbf{S} \overline{\Theta})$) of the various transformations across training steps with the 200M model. The projections with lowest reconstruction error (measured either by L2 error or cosine similarity with the unprojected stochastic gradient) do not give the best downstream performance (see Tab. 2).

## D. Experimental Setup

The details of the two architectures we consider are shown in Tab. 5. We also include a discussion on gradient accumulation.

**Gradient accumulation.** The experimental setup in the original GaLore paper did not perform gradient accumulation, which meant that the maximum sequence length had to be short enough (e.g., 256) such that a single batch could contain a large-enough number of sequences for accurate gradient estimation. Our experiments are instead conducted in the standard setting where we assume gradients are accumulated across multiple microbatches. In this case, the reparameterization of GaLore as LoRA has the additional benefit of straightforwardly allowing for gradient accumulation in the lower-dimensional space. Concretely, the most straightforward implementation of GaLore[17] will lead to gradient accumulation in the original parameter space, which would consume substantially more memory. In contrast, in the LoRA formulation the gradients are accumulated *after* applying the gradient transformation, providing substantial memory savings without any additional code.[18]

|  | 200M | 1.3B |
|---|---|---|
| Layers | 12 | 24 |
| Heads | 16 | 16 |
| Embed. dim. | 1024 | 2048 |
| Intermediate dim. | 2816 | 5472 |
| Head dim. | 64 | 128 |
| Query groups | 16 | 16 |
| Batch size | 0.5M | 1M |
| Warmup tokens | 0.5B | 1B |
| Total tokens | 5B | 10B |

**Table 5:** Description of the two architectural settings we consider for our experiments: a 200M setting which we conduct most of our analyses and ablations on, and a 1.3B setting which we use to evaluate our techniques in more realistic, large-scale setting.

---

[15]Note that since the right hand side lies in $\text{range}(\mathbf{S}^\top)$, this linear system will have a solution.

[16]In the distributed case, this might not incur any additional cost, since $\Theta^{(0)}$ would already need to be stored. But this would require solving a linear system on every iteration.

[17]E.g., the official implementation in https://github.com/jiaweizzhao/GaLore.

[18]One can implement this optimization in the GaLore form, but this requires additional code. The issue is that most deep learning frameworks will compute the gradients of a parameterized function, and the user then separately passes these as input to an optimizer. If gradients are only transformed in the optimizer, then the automatic differentiation module cannot figure out that only the smaller transformed gradient is needed, and not the full gradient. We suspect that a sufficiently good compiler should in principle recover this optimization if one ensures that entire training steps (viz., all the gradient accumulation steps and optimizer step) are compiled jointly.

