# OpenReview forum: "On the Duality between Gradient Transformations and Adapters"
_ICML.cc/2025/Conference — ICML 2025 poster_

### Official Review · Reviewer_u2rQ · 2025-03-10

**Overall Recommendation:** 5

**Summary:**

The authors derive an equivalence between LoRA / adapter training methods, which optimise a low rank addition to weight matrices to reduce optimiser memory usage, and methods that project gradients themselves down to low rank (such as Galore), also to reduce optimiser memory usage. Their result generalises more specific cases that have been shown in recent works. They then investigate the practical insights that might be gained from this observation, performing various LLM pretraining experiments with a range of adapter / gradient projection techniques, and propose a simple modification in the distributed training scenario motivated by the correspondence they prove, which improves performance.

### Update after rebuttals
I have maintained my score of "Strong accept" - please see rebuttal comment

**Claims And Evidence:**

- The primary claim of the paper is that they prove that adapter methods are equivalent to gradient low-rank projection methods, which is true. Their theorem covers a general case that applies to most of the recent works such as lora and galore.
- Their secondary claim of utilising this theory to develop a better algorithm in the distributed training case is also backed up by good experiments, with relatively large models (1B).

**Essential References Not Discussed:**

None that I am aware of.

**Experimental Designs Or Analyses:**

The experimental setup is sound, pretraining LLama models on large text datasets with LoRA adapters from scratch. I was slightly confused by what the authors meant by the "Transformer++" architecture, as I could not find this in the cited LLama paper.

**Methods And Evaluation Criteria:**

- The proposed theorem and corresponding proof are relevant to the well-known lora and galore algorithms. In particular, they assume that the gradients are projected down to a low-rank subspace using a matrix $\mathbf S$, and the optimiser steps are then projected back up to the full space using its transpose $\mathbf S^T$. This is the appropriate setup when $\mathbf S$ has orthonormal rows / columns, or in the case where $\mathbb E[\mathbf S \mathbf S^T]=\mathbf I$. I would also be interested to know whether a similar equivalence can be proved in the case where you project back up using the Moore-Penrose pseudo-inverse, which is equivalent to $\mathbf S^T$ when $\mathbf S$ is orthogonal but not in general. However, this case might not be of practical relevance anyway, as the Moore-Penrose inverse would be an expensive step to compute as part of an optimisation loop.
- The comparison between various adapter and gradient transformation methods in Table 2 was interesting, though I wasn't sure how this was directly relevant to the equivalence they proved.
- The proposed modification to training in the distributed setting definitely sounds clever, since they use different projections for the different workers to ensure that the overall parameter updates are roughly full rank. However, this seems to be specifically in the one-sided LoRA setting, and I'm not sure whether the problems they fix are relevant in the more usual two-sided LoRA setting.

**Other Comments Or Suggestions:**

- Please define the notation $\text{vec}^{-1}$ first used on line 195.
- I didn't fully understand the points about why different schemes need different persisted matrices in Table 1. Slightly more explanation would be helpful.
- I also found the discussion around Table 4 slightly confusing. It seems to show that in the baseline, having more workers is beneficial, but in the lora experiments, having more workers is detrimental?
- It would be useful to clarify what the metrics are in all table captions (I think they are all test perplexity?).
- In proof of theorem 1, I found equation 5 hard to follow, but it was just the chain rule. It might be worth slightly rewriting this small segment to make it clearer.
- Typo: in line 759, "imilarly"

**Other Strengths And Weaknesses:**

### Other strengths
- I think the proposed equivalence is very useful, as many PEFT schemes like this have been proposed, and it turns out that in many cases they are equivalent.
### Other weaknesses
- The experiments in the paper, other than the distributed training setting, do not seem entirely relevant to the main point of the paper. However, since the main point of the paper is the proved equivalence, the experiments are of less importance.

**Questions For Authors:**

- What is the Transformer++ architecture? Perhaps I have missed something, but I cannot find it in Touvron et al 2023.
- Is it possible to prove a similar equivalence where you use the moore-penrose pseudo-inverse to project back up? (I.e. your low rank projection isn't necessarily orthogonal).
- (not very important but I was curious): Do you have an explanation for why the SVD curve in Figure 1 plateaus over time?

**Relation To Broader Scientific Literature:**

The discussed schemes of lora, galore, relora etc. are all very relevant and discussed methods in the current PEFT literature. As mentioned in this paper, previous papers have proved equivalences between some of these schemes, such as one-sided lora and galore, and this paper serves to generalise these proofs to a much wider setting that includes many of these schemes.

**Theoretical Claims:**

I checked all the proofs in Appendix A. Theorem 1's proof is a neat proof by induction that takes less than a page. The other two proofs, for proposition 2 and corollary 3, are straightforward applications of Theorem 1 using Kronecker factored matrices, but are still useful to have written in full.

---

> ### Author Rebuttal · Authors · 2025-04-01
>
> Thank you for your review! We address your questions below:
>
> ### Key points
>
> > Is it possible to prove a similar equivalence where you use the moore-penrose pseudo-inverse to project back up?
>
> Great question! It is possible that an equivalence exists if one is not using a LoRA adapter. For example, one could imagine defining a PyTorch primitive function whose forward pass behaves like a regular (one-sided) LoRA adapter $S^\top A$, but whose backward pass computes the gradient of $A$ using the Moore-Penrose inverse instead of $S^\top$. In principle this should be equivalent to training with GaLore (with a Moore-Penrose up-projection) in the optimization trajectory sense, though we are not quite sure what the mathematical interpretation of this would be.
>
> > The comparison between various adapter and gradient transformation methods in Table 2 was interesting, though I wasn't sure how this was directly relevant to the equivalence they proved.
>
> The main aim of Table 2 is to compare (empirically) different choices of $S$, specifically to understand the trade-off between test PPL and estimated memory usage. The results suggest that Rademacher matrices offer better memory efficiency at a slight PPL penalty, and randomized semi-orthogonal matrices offer an even smaller penalty, though perhaps not a memory improvement (since it is not as clear how one could rematerialize them efficiently).
>
> > The proposed modification to training in the distributed setting definitely sounds clever [...] however, this seems to be specifically in the one-sided LoRA setting [...]
>
> Yes, you are correct---this result is specific to the one-sided LoRA setting. We focused on this setting because this is what the theorems we prove focus on. That said, we also tried running a full two-sided distributed training setting (which would be equivalent to distributed ReLoRA/LTE; 20.97 PPL [200M] and 13.72 PPL [1.3B]) and found it to underperform the one-sided LoRA with worker-aware initialization. We will include these results in the next version of the paper.
>
> > Do you have an explanation for why the SVD curve in Figure 1 plateaus over time?
>
> Great question! We suspect that towards the end of training, since the learning rate has annealed, the model is actually shifting very little in parameter space. This means that the gradient is also shifting fairly little. This means that our SVD-based gradient estimator, which is optimal (in the Frobenius error sense) when applied to the gradient the SVD was computed on, remains fairly optimal.
>
> > I was slightly confused by what the authors meant by the "Transformer++" architecture
>
> Thanks for pointing this out. This part is unclear, and we will fix this in the next version of the paper. By Transformer++ all we mean is a Transformer with the changes made in the Llama architecture, e.g., RMSNorm instead of LayerNorm, SwiGLU activations, rotary embeddings (some papers use this term).
>
> ### Other comments/suggestions
>
> Thank you for your comments here—we will address them all in the next version of the paper. We also include a brief explanation/clarification where appropriate:
>
> > Please define the notation vec^-1  first used on line 195.
>
> If $\text{vec}(\cdot)$ sends a matrix to its vectorized form (i.e., we stacked up the columns), then $\text{vec}^{-1}(\cdot)$ takes a big stacked vector and unstacks it into a matrix of the right size.
>
> > I didn't fully understand the points about why different schemes need different persisted matrices in Table 1
>
> This point is related to your earlier question about Table 2. Table 1 is trying to be explicit about the cases where it should be possible to just persist the seed that generates a matrix, and regenerate it on-the-fly as needed. This means additional memory is saved since we can get away with just storing the seed for that matrix (~4 bytes), as opposed to storing the matrix itself.
>
> > I also found the discussion around Table 4 slightly confusing. It seems to show that in the baseline, having more workers is beneficial, but in the lora experiments, having more workers is detrimental?
>
> Great question---In Table 4 we simultaneously vary the number of workers and the rank of the gradient transformation/LoRA adapter, except in the case of the baseline, where we only vary the number of workers. Since we allow all workers to train for the same number of tokens, this means that for the baseline, increasing the number of workers increases the effective number of total tokens we train on *without* any penalty (i.e., reducing the rank). So the baseline will always benefit from more workers without any real negative effects, whereas the other approaches will suffer from the rank being decreased as the number of workers is increased. We mention this briefly in the caption, but we will make sure to make this clearer.
>
> > It would be useful to clarify what the metrics are in all table captions (I think they are all test perplexity?).
>
> That’s right, they are all test perplexity.

---

> > ### Comment · Reviewer_u2rQ · 2025-04-02
> >
> > Thank you for your response. I have also read the other reviews and responses, and would like to maintain my score of "Strong Accept". I think this is an excellent piece of work, and the equivalence proved is interesting and should help combine previously separate research directions.

---

### Official Review · Reviewer_kUHz · 2025-03-10

**Overall Recommendation:** 4

**Summary:**

This paper studied the connection between weight-efficient optimization and gradient-efficient optimization of transformers, and found that optimizing a model in a compressed gradient space is equivalent to optimizing just an additive low-rank adapter. Through theoretical analysis and empirical study, the authors showed the equivalence between those two lines of efficient optimizations.

**Claims And Evidence:**

Yes

**Essential References Not Discussed:**

No

**Experimental Designs Or Analyses:**

Yes

**Methods And Evaluation Criteria:**

Yes

**Other Comments Or Suggestions:**

No

**Other Strengths And Weaknesses:**

### Strengths:
- The writing is clear and easy to follow.
- The derivations make sense to me.

### Weaknesses:
- The empirical study is not clear enough to me, regarding supporting the previous claim about the equivalence between weight-efficient optimization and gradient-efficient optimization. Table 2 is more like a performance comparison instead of showing any connections between different methods. Besides, the analysis in this table is also ambiguous.

**Questions For Authors:**

1. Could you further explain your findings from experiments in Sec 4.1? Like how do these results support the equivalence between weight-efficient optimization and gradient-efficient optimization, and how is this equivalence used to build more efficient training methods?

**Relation To Broader Scientific Literature:**

This paper is a revisit to previous memory-efficient training methods [1, 2, 3] for transformers. It aims to discover the connections and build equivalence between them, both theoretically and empirically.

[1] Zhao, Jiawei, et al. "GaLore: Memory-Efficient LLM Training by Gradient Low-Rank Projection." International Conference on Machine Learning. PMLR, 2024.

[2] Hu, Edward J., et al. "LoRA: Low-Rank Adaptation of Large Language Models." International Conference on Learning Representations. 2022.

[3] Lialin, Vladislav, et al. "ReLoRA: High-Rank Training Through Low-Rank Updates." International Conference on Learning Representations. 2024.

**Theoretical Claims:**

Yes. The theoretical derivations are solid to me.

---

> ### Author Rebuttal · Authors · 2025-04-01
>
> Thank you for your response! We will break down your question into parts:
>
> > Could you further explain your findings from experiments in Sec 4.1?
>
> The key goal of Sec 4.1 is to understand the interplay between the choice of gradient transformation S and overall performance. One important aspect of this interplay is that some transformations are expensive to update (e.g., the SVD transformation works well according to the GaLore paper, but requires running an SVD which is expensive), and others take more memory to store (e.g., the SVD transformation requires you to persist that matrix in memory, but if you are using random Gaussian matrices you just need to store a seed that you will use to regenerate this matrix—this means the whole matrix can be compressed into a handful of bytes).
>
> Table 1 tries to summarize these trade-offs conceptually. For each gradient transformation method, we describe what the parameters (of linear layers) in the model would look like using this approach, which of these parameters would be trained/frozen, and which of these need to be persisted in memory.
>
> Table 2 shows the empirical results (in terms of PPL on a held-out subset of SlimPajama) of training 200M and 1B using each of these methods; since the LoRA formulation is amenable to quantizing the frozen weight, we also include for each method the results using INT8 quantization and NF4 quantization. Beyond the test PPL, we also report the estimated memory usage of each approach.
>
> The most important point of Table 2 is that, at a minor loss of PPL (i.e., +0.1-0.3 PPL), one can use other transformations that offer memory efficiency (e.g., Rademacher gradient transformations). The other takeaways from the table are that (i) while two-sided gradient transformations may be intuitively nice, they don’t seem to work as well, and (ii) INT8 quantization gives you memory savings without any meaningful PPL degradation, but NF4 starts to incur a non-trivial PPL penalty.
>
> But we agree that section 4.1 could be explained better and will make sure to expand on the above points in the next iteration of the paper.
>
>
> > The empirical study is not clear enough to me, regarding supporting the previous claim about the equivalence between weight-efficient optimization and gradient-efficient optimization. Table 2 is more like a performance comparison instead of showing any connections between different methods
> > [...] how do these results support the equivalence between weight-efficient optimization and gradient-efficient optimization
>
> Since we prove that there is a mathematical equivalence between training with gradient transformations and training with a weight transformation (aka., a linear, one-sided adapter), the goal of our experiments is *not* to empirically verify this equivalence (since this would be redundant), but instead to leverage it by exploring, e.g., the interplay between choice of gradient transformation and empirical language modeling performance. That said, we will add a small scale experiment in the appendix where this equivalence is validated empirically, by showing that the loss curves of two toy models remain the same during training.
>
> > [...] and how is this equivalence used to build more efficient training methods?
>
> Beyond the new gradient transformations we propose in Table 2, our biggest contribution in terms of more efficient training methods is in combining gradient transformations with distributed training. In particular, in Table 3 we show that making different workers have different gradient transformations *and* making these transformations pairwise orthogonal (i.e., the distributed random method) is better than having them be orthogonal (but not pairwise orthogonal) or having them be the same across workers. In Table 4, we also study the effect of varying the number of workers, and find that the benefits of the distributed random method become more pronounced as the model is partitioned across more and more workers (each training a smaller slice of the model).

---

### Official Review · Reviewer_oyUd · 2025-03-20

**Overall Recommendation:** 3

**Summary:**

This paper explores a memory-efficient approach to neural network optimization by mapping gradients into a lower-dimensional space, reducing the memory overhead of both gradient accumulation and optimizer states. After performing updates in this reduced space, parameters are returned to the original space via the transpose of the linear transformation. The authors demonstrate that optimizing in this transformed gradient space is equivalent to reparameterizing the model through a linear adapter—which additively modifies the parameters—and training only the adapter. By employing a Kronecker-factorization of this linear map, the paper further clarifies the connection between GaLore and LoRA, showing they coincide (for one-sided LoRA) under this unified perspective.

**Claims And Evidence:**

Yes.

**Essential References Not Discussed:**

-

**Experimental Designs Or Analyses:**

Yes, and in comparison with the related works different model sizes need to be investigated to show if the method works.

**Methods And Evaluation Criteria:**

The proposed method is interesting but more datasets are needed to be explored.

**Other Comments Or Suggestions:**

-

**Other Strengths And Weaknesses:**

The paper has comprehensive related works.
It is also well written and well organized

**Questions For Authors:**

I wrote more comments above but  how the convergence or the collapse of the method will be with this method?

**Relation To Broader Scientific Literature:**

The idea is interesting and could be practical in terms of scale.

**Theoretical Claims:**

Yes, and they seemed correct to me.

---

> ### Author Rebuttal · Authors · 2025-04-01
>
> Thank you for your review!
>
> Our main objective in this paper was to explore pretraining, and thus we selected a pretraining dataset that matches the distribution of pretraining data (SlimPajama, which is based on RedPajama, that tried to match the pretraining data used by Llama, one of the most successful open-source LLMs).
>
> In terms of model size, because of compute constraints in an academic setting, we scaled as large as we could go (1B). As much as we’d love to scale further, we cannot do a pretraining run of a larger scale on our resources.
>
> > I wrote more comments above but how the convergence or the collapse of the method will be with this method?
>
> Besides the standard convergence results that apply to ML models of this scale, we note that, due to the nature of our equivalence, the theorems in the GaLore paper should still apply (subject to those theorems’ conditions, e.g., any restrictions on the choice of projection). Empirically, we find that all approaches we consider have similar convergence behavior.

---

### Official Review · Reviewer_KsUQ · 2025-03-20

**Overall Recommendation:** 2

**Summary:**

This paper investigates the duality between linear gradient transformations and adapter-based reparameterizations in memory-efficient LLM training. In essence, it shows that applying linear transformations to gradients (as in GaLore) is equivalent to reparameterizing the model via adapters (like one-sided LoRA). This connection is interesting, as it unifies several existing methods and could potentially inspire new techniques. However, while the comprehensive summary and explicit unification of existing methods are very helpful, its conceptual insight is somewhat already quite clear from the literature, the contributions beyond what is already known appear incremental.

**Claims And Evidence:**

The paper claims that its generalized duality result not only recovers the known equivalence between GaLore and one-sided LoRA but also extends to more general settings. Although the authors provide a comprehensive summary of various optimizers and their relationships (as seen in Table 1), the result itself is rather straightforward. In the empirical evaluation, the best-performing method on the 1B model is still the gradient SVD adapter version of GaLore, and when compared to LoQT, the only difference is a slight change in quantization precision. These findings, while interesting, do not convincingly demonstrate fundamentally new methods that offer a significant improvement over existing ones.

**Essential References Not Discussed:**

N/A

**Experimental Designs Or Analyses:**

It seems that in the pretraining experiment, the authors mostly follows the setups (same number of tokens, etc) of the Galore paper (perhaps with different context length but this is not specified in the paper). However, according to the 1B experiment all baselines uniformly performs much better than the original setup. I would appreciate a comparison under the same setup to give readers a sense of how well these methods performs. Moreover, while ReLoRA is presented as a strong baseline in the paper, its performance in prior benchmarks has generally been weak. This discrepancy raises questions about whether the reported improvements are robust and representative.

**Methods And Evaluation Criteria:**

There are a few minor limitations. See "Claims And Evidence" and "Experimental Designs Or Analyses"

**Other Comments Or Suggestions:**

N/A

**Other Strengths And Weaknesses:**

A notable strength of this work is its focus on an important direction—understanding the equivalence between optimizers and weight-space transformations in memory-efficient training. This is a valuable perspective that could potentially lead to new insights. On the downside, the technical contributions are incremental. The generalization of the duality between Galore and LoRA is quite straightforward, and more importantly, did not lead to new research insights (in terms of predicting new methods beyond what is known). The empirical results, particularly on the largest benchmark, do not convincingly demonstrate the intended claims.

**Questions For Authors:**

- How do you justify the experimental setup, particularly for the 1B model, where all baselines perform uniformly better than reported in previous work? Would a side-by-side comparison under the exact same setup offer different insights?

- Given that ReLoRA is shown as a strong baseline in your paper, how do you reconcile this with prior benchmarks where ReLoRA has performed poorly?

- Can you comment on whether the duality result leads to any fundamentally new methods or improvements beyond providing a unifying view of existing techniques?

**Relation To Broader Scientific Literature:**

N/A

**Theoretical Claims:**

The paper’s theoretical result—the equivalence between gradient transformations and adapter reparameterizations—is elegant but not surprising. Generalizing this duality appears to be a natural extension of previous work, and it does not suggest fundamentally new techniques. In particular, the derivation does not yield any novel optimizer that outperforms existing ones, but rather shows that the methods are different views of the same underlying mechanism.

---

> ### Author Rebuttal · Authors · 2025-04-01
>
> Thank you for your review! We identify three key criticisms pointed out in your review, and address them below
>
> ### 1: Conceptual insights in the duality derivation are simple
>
> Indeed, while the essence of our derivation was observed previously in the literature, our derivation is more general; it considers a linear adapter applied to any parameter vector and it takes into account optimizer choice (previous works were only showing the equivalence for parameter matrices of linear layers and SGD, a non-stateful optimizer). This view is productive: it allows us to establish a connection between Kronecker-factored gradient transformations and recent two-sided variants of LoRA adapters which have been found to perform well for parameter-efficient finetuning, in particular MoRA, PMSS, and LoRA-XS. This is to the best of our knowledge novel in the literature. We think that this more general result is a worthwhile contribution to the literature.
>
> ### 2: The empirical contributions borne out by the insights of the derivation are incremental.
>
> A key criticism seems to be that despite proposing other gradient transformations (e.g., Rademacher, randomized semi-orthogonal), the SVD transformation (i.e., GaLore) is still the highest performing transformation in Table 2 in terms of PPL in the 1B case. However, in Table 2 we see that the SVD transformation underforms the proposed approaches in the more memory-efficient NF4 quantization (i.e., QLoRA) setting: the Rademacher and Random-Orthogonal projections in this setting perform nontrivially better. Furthermore, in the 1B setting without quantization (or with INT8 quantization), the gap between the SVD and some of the other approaches is fairly modest (e.g., 0.1-0.3 PPL), and it is important to take into account that some of these are more memory efficient than the SVD transformation (see memory column in Table 2 for, e.g., Rademacher transformations).
>
> We also kindly note that a large portion of our paper was dedicated to the application of the gradient transformation–adapter duality to distributed training, though it seems this wasn’t mentioned in the review. In particular, we show that choosing the transformations so they are worker-aware leads to better results than making them randomly different across workers or making them the same across workers.
>
> Finally, we want to emphasize that the goal of the paper wasn’t necessarily to achieve state-of-the-art memory-efficient pretraining results, but more to give a flavor for how our duality can unify existing approaches and lead to new techniques.
>
>
> ### 3: Discrepancy between our results and those in the GaLore paper
>
> First, we’d like to stress that our setup is different from the GaLore paper in many respects, so it is hard to do a direct comparison. For example, we use a different dataset (SlimPajama instead of C4), we use gradient accumulation, our maximum sequence lengths are substantially higher (2K instead of 256), and our optimizer hyperparameters are not the default Adam hyperparameters, but instead are more consistent with values used in more recent, large LLM training runs (we use the ones for Llama-2 and OLMO).
>
> More importantly, our decision to use a different setup was deliberate: Since we focus on pretraining, we want to be in the setting that most closely resembles recent LLM training runs. Therefore, we do not see the discrepancies between our results and those in the GaLore paper as a reflection that our setup is suboptimal; rather, we see our results, in part, as an attempt to apply GaLore to a different setting. We also note that such nontrivial differences in performances due to different training setups is common: for example, the Flora paper’s replication of GaLore (in Table 6 of the appendix) also shows substantially different results.
>
> ## Answers to individual questions
>
> 1. See point 3.
>
> 2. This is a great question—we were also puzzled by this. Here, we actually think it is worthwhile taking a step back and looking at the results and models more broadly. Mathematically (from our duality derivation), it is clear that GaLore (and thus one-sided LoRA training) partitions the parameter space and trains only along a subspace of it. This reduces the effective number of parameters in our model, which would usually lead to a performance hit. Further, from our derivation, ReLoRA can be seen as GaLore where the projection is learnt by the optimizer, so unless this small number of new parameters introduce a lot more instability, we arguably shouldn’t expect the results to change much. Both of these observations manifest in our results. From this lens, we think our results are arguably what we’d a priori expect. While we cannot be sure why previous work doesn’t also show this same trend, but we suspect that this may have to do with the differences in the experimental setup (see point 3 above).
>
> 3. See point 2.

---

### Decision · Program_Chairs · 2025-05-01

**Decision:**

Accept (poster)

**Comment:**

This paper identifies and explores a duality between linear gradient transformations like GaLore and certain adapter-based reparameterizations like one-sided LoRA. The reviewers found this connection quite interesting and useful as it unifies several existing techniques and could open new ways of thinking about memory-efficient LLM training.

While the reviewers were generally positive about the main contributions, some reviewers found the experimentation to be unclear or unconvincing, and others found the connection itself to be elegant but unsurprising. Given that the author response clarified many aspects of the experiments, and that in any case the experimentation is not the main contribution of the paper, I find these concerns have been adequately addressed. In addition, even if the connection is not surprising, establishing it concretely as this paper does is an important contribution and I believe the ICML community will find the results quite interesting; therefore, I recommend acceptance.